# The Moderating Effects of Self-Referencing and Relational-Interdependent Self-Construal in Anti-Smoking Advertising for Adolescents

**DOI:** 10.3390/ijerph17228481

**Published:** 2020-11-16

**Authors:** Hsiang-Ming Lee, Ya-Hui Hsu, Tsai Chen

**Affiliations:** 1Department of Health and Welfare, University of Taipei, Taipei 11153, Taiwan; hmlee@utaipei.edu.tw; 2Department of Business Administration, Ming Chuan University, Taipei 11103, Taiwan; 3Department of Radio, Television and Film, Shih Hsin University, Taipei 11604, Taiwan; chenctng@gmail.com

**Keywords:** relational-interdependent self-construal, self-reference, advertising appeal, adolescents’ smoking intention

## Abstract

The tobacco epidemic is one of the most serious public health issues in the world. Tobacco use starts and becomes established primarily during adolescence, and nearly 9 out of 10 cigarette smokers first tried smoking by age 18, with 99% first trying by age 26. This study employed a 2 (advertising appeal: emotional vs. rational) by 2 (self-referencing: analytical vs. narrative) factorial design in Study 1; and a 2 (relational-interdependent self-construal: high and low) by 3 (social relational cue: self, friend, and family) factorial design in Study 2. The behavior intention of anti-smoking acted as the measured dependent variable. Samples of 192 (Study 1) and 222 (Study 2) were collected from one of the biggest high schools in northern Taiwan. The results showed advertising appeal and self-referencing had a significant interaction effect on behavior intention (*p* = 0.040). The results also showed rational appealing advertising is suitable for analytical self-referencing (*p* = 0.022) and emotional appealing advertising is suitable for narrative self-referencing (*p* = 0.067). However, the social relationship cue and relational-interdependent self-construal had no significant interaction effect on behavior intention, and only relational-interdependent self-construal significantly affected behavior intention (*p* < 0.001). Regardless of whether the relational-interdependent self-construal is high or low, when the anti-smoking advertising is from the family perspective to persuade adolescents not to smoke, both influence the adolescent more than the other two social relationship cues (self and friend).

## 1. Introduction

The tobacco epidemic is one of the most serious public health issues in the world, causing more than 7 million deaths each year. More than 6 million of these deaths are the result of direct smoking, while approximately 890,000 are the result of non-smokers’ exposure to secondhand smoke [1]. Based on the U.S. Centers for Disease Control and Prevention (CDC) reports, tobacco use is started and established primarily during adolescence, and nearly 9 out of 10 cigarette smokers first tried smoking by age 18, with 99% first trying by age 26. Thus, preventing the use of tobacco by the youth age group is critical to stop another generation of adults from smoking and suffering from smoking-related death and disease [2].

Over the past few decades, media campaigns have attempted to influence various health behaviors in mass populations [3]. Of the common behaviors targeted by media campaigns, smoking cessation has perhaps been the most prominent [4,5]. Many studies examined the effect of anti-smoking messages from different perspectives, such as nonsmokers’ responses to anti-smoking public service announcements [6] and graphic visual health warnings [7], or used content analysis to look into the theories and messages from different countries [8,9,10]. Most of the research used gain- or loss-framed messages [11] or appeals to emotion, such as fear of death [12] or anger [6], to examine the anti-smoking effect. However, some research (e.g., Cooper, Goldenberg, and Arndt [13]) suggests explicit reminders of death often have no significant effects on health-related attitudes and behaviors, including quitting smoking, increasing exercise, protecting one’s health, and being optimistic about the outcomes of his or her health risk assessment (health optimism); rather, the impact is moderated by dispositional or situational variables. Therefore, we recommend that advertisers persuade the audience of a product’s value or healthy behavior by encouraging them to relate the product or healthy behavior either to themselves or to their own experience. Meyers-Levy and Peracchio [14] suggested this process of relating to oneself, known as self-referencing, can benefit ad persuasion. The positive effect of self-referencing on the ad occurs regardless of the potentially detrimental effects of self-referencing on learning about the ad subject’s features and attitude [15]. Ducoffe [16] defined the advertising effect as the effects the advertisers and audience communicate to conduct potential transactions via advertising messages, which also provides the basis for judging advertising success. The advertising effect means after the audience watches the advertisement, their behavioral decision is affected by the advertisement [17]. Many studies have used self-referencing to examine the advertising effect, but research has seldom utilized self-referencing to understand the effect of anti-smoking advertisements. In an advertising campaign, an important issue in developing advertisements is determining the type of information and the benefits of communication [18]. Selecting the type of advertising appeal used for each target group is an important strategy [19]. Thus, one purpose of this study is to use self-referencing and advertising appeal to examine the effect of anti-smoking advertisements.

Prior studies suggested peers and parents play a role in influencing adolescents’ decisions concerning smoking [20]. According to the relational-interdependent self-construal (RISC) aspect, which is defined as the tendency to think for oneself in terms of relationships with close others [21], RISC significantly affects cognition, emotion, motivation, and social behaviors related to the individuals’ relationships [22]. People with a high level of RISC maintain harmonious interactions with others [23]. Most of the RISC research concerns the effects of friends [24,25], but the effect of RISC from the family perspective or other social relationship cues has seldom been investigated. In addition, Martin, Lee, and Yang [26] suggested RISC and self-referencing can make social marketing advertising more effective. Thus, the second purpose of the present study is to demonstrate the effects of RISC, social relationship cues, and self-referencing on anti-smoking advertising.

The content provided by the advertisement’s message, the way the message is presented, and the characteristics of the recipients themselves all affect adolescents’ acceptance of anti-smoking advertisements. Both the theoretical and practical contributions of this research are as follows: (1) Drawing on multidisciplinary concepts taken from sociology and psychology, this paper empirically tested how anti-smoking ad appeals, self-referencing, and relational-interdependent self-construal affect adolescents’ anti-smoking intention. (2) To our knowledge, there is no research discussing the effect of self-referencing and relational-interdependent self-construal on anti-smoking advertising. This research examined these psychology variables to fill this research gap. (3) Previous advertising studies on youth smoking are sparse, and most of them used survey methods to explore the advertising effects, e.g., Trumbo and Kim [27] and Shadel et al. [28]. This research used experimental methods to explore the causal relationships among these variables, which is irreplaceable by the survey method.

## 2. Literature Review

### 2.1. Anti-Smoking Advertising Appeal

When advertisers seek greater effectiveness in communication, they must more carefully consider the selection of the type of advertising appeals used for each target [19]. Advertising appeals are used to attract the audience’s attention, as well as influence their attitudes and emotions regarding the product, service, or idea [17,29]. If an audience has a favorable feeling toward an advertisement, they are more likely to accept and even “enjoy” the ad message [30].

Trumbo and Kim [27] used the survey method to examine whether video advertisement appeals and beliefs about the addictiveness of e-cigarettes affect college students’ uptake of e-cigarettes. They found different advertising appeals influence the uptake of e-cigarettes. Shadel et al. [28] also used the survey method to examine the effects of tobacco advertising on five classes of tobacco product (electronic cigarettes, hookah, cigars, cigarillos, and smokeless tobacco) on tobacco use among youth experiencing homelessness. They found advertising appeal was positively associated with future intentions to repeat the use of electronic cigarettes and hookah. Lin and Zhang [31] focused on the key role of anti-smoking messaging (hedonic vs. utilitarian) to mitigate precipitation’s impact on cigarette consumption. Amonini et al. [32] found the “shame appeal” television advertisement resonated with smokers and encouraged quitting/reducing behaviors. However, advertisers can create different types of appeal by providing different types of information [18], and they have developed two important message appeal strategies: rational (or functional) and emotional appeals [33]. Advertising with rational appeal is an advertisement stimulus supplying factual information about the product, service, or idea [34,35]. Advertising with rational appeal often focuses on the utilitarian benefits, including messages regarding quality, economy, performance, value, and reliability [33,36]. On the other hand, emotional appeal in advertising aims at evoking feelings or the emotions of the audience [37]. Provoking positive emotions (e.g., love, pride, humor, and joy) or negative emotions (e.g., fear and guilt) may stimulate specific responses and behaviors [33]. The present study examined the effect of emotional and rational appeal on anti-smoking advertising.

### 2.2. Self-Referencing

Self-referencing is a processing strategy individuals use to process information by relating a message such as an advertisement to their own self-construct [26,38]. In cognitive psychology, self-referencing is conceptualized as the cognitive processes individuals associate with incoming information, involving information previously stored in the memory to give the information new meaning [39]. Some psychology studies demonstrated self-referencing enhances learning and the recall of information [40] in the advertising field, as well as playing a role in persuasion [39]. A related study about advertising also found higher levels of cognitive self-referencing and positive effects when the ad image was congruent with their ideal self-schemata than when it was not [41].

Self-referencing is a multidimensional construct and some research has proposed the classification of self-referencing. Based on the time frame, Krishnamurthy and Sujan [42] divided self-referencing processes into retrospective and anticipatory self-referencing. Retrospective self-referencing is self-referencing processed by referring to autobiographical experiences and events from one’s past. Anticipatory self-referencing is self-referencing processed by referring to imagined experiences and events from one’s future [42]. Escalas [40] proposed two different kinds of self-referencing: narrative and analytical self-referencing. Narrative self-referencing processing affects persuasion through transportation, which is defined as immersion in a text [43]. A narrative structure comprises two critical components: chronology and causality. Chronology means narrative events occur over time so viewers can perceive the events’ beginning, progress, and ending according to their time flow. Causality then connects the story’s events to causal inferences [44]. The literature suggests that, by using drama or storytelling, narrative advertising can captivate and mesmerize its audience through the dramatic unfolding of causally related events in the form of storytelling or drama, “transporting” the viewer to the narrative world [45]. That is, when individuals become immersed in a drama or narrative story and begin to experience the characters’ world vicariously, they are “transported” and “hooked” in the narrative world [45,46]. On the other hand, analytical self-referencing persuades through dual cognitive response processes (e.g., Elaboration Likelihood Model; ELM). These traditional elaboration-based persuasion models state that when the ad arguments are strong, self-referencing facilitates the elaboration of incoming information, improving message recall and ad and brand attitudes [40]. In analytical self-referencing studies, the ad text is written in the second person (“you”) with a few requests to recall generic, repeated incidents [38,40], or with a photo taken from the consumers’ perspective [14,40]. The present study used narrative and analytical self-referencing to survey the effect of anti-smoking advertising.

In the research on self-referencing, Chang [41] found consumer self-referencing influenced brand attitudes via its influence on ad attitudes. Ching et al. [45] found self-referencing substantially affects transportation in forming product attitudes as well as positively affecting product attitude. In addition, Escalas [40] also proposed that the transportation effect is more likely to occur in high-self-referencing consumers.

### 2.3. The Interaction Effect between Self-Referencing and Advertising Appeals

Based on the transportation theory, when an individual is absorbed into a story or transported into a narrative world, he or she may display the effect of the story in their real-world beliefs [43]. A transported reader might experience a strong motivation and emotion, even though he/she knows the event in the story is not real [43,47]. Ardelet, Slavich, and de Kerviler [48] proposed consumers naturally engage in narrative (rather than analytic) processing in a store when encountering luxury products because of their subjective and emotional dimensions. Fabrigar and Petty [49] also showed emotional appeal (vs. cognitive appeal) in an ad is more persuasive for the audience whose existing attitudes are based on an affective (vs. cognitive) process. Rational appeals focus on the practical, functional, or utilitarian benefit coming from the use of the product [50,51]. Consumers prefer rational advertising appeals because they can provide information that clearly explains the difference between the advertised brand competitors [51,52]. From the above, according to the ELM model, if the advertising is aimed at evoking feelings or emotions in participants, the receivers with more narrative self-referencing will be more receptive to the anti-smoking ad messages and increase their behavioral intention of anti-smoking. On the other hand, if the advertising provides more information about how smoking will hurt their health and economy, the receivers with more analytical self-referencing will be more receptive to the anti-smoking ad messages and increase their behavioral intention of anti-smoking. From the above, we propose the following hypotheses:

**Hypothesis** **1.**For emotional appeal advertising, narrative self-referencing leads to better behavioral intention of anti-smoking than analytical self-referencing.

**Hypothesis** **2.**For rational appeal advertising, analytical self-referencing leads to better behavioral intention of anti-smoking than narrative self-referencing.

### 2.4. Relational-Interdependent Self-Construal

Self-construal refers to an individual’s sense of self in relation to others, and two primary types of self-construal have been identified: independent and interdependent [53,54]. Independent self-construction is defined as a limited overall stable self, which is separated from the social environment [55]. However, interdependent self-construction is understood as a larger whole self in terms of connections to important others and groups [56]. That is, the individual adopts an interdependent self-construal, and their self-concept depends largely on their capacity to establish and maintain their connection to a broader social entity [57,58]. In addition, RISC relates to an individual’s dyadic relationships (e.g., spouse or close friend) instead of the individual’s relationship with generalized others (e.g., same ethnicity) [59]. Hence, compared to low RISC, high RISC encourages people to promote and maintain harmonious interactions with others [60] and when making important decisions, high RISC will consider the needs and wishes of others [56].

Some advertising research concerning construal and persuasion found that people with a salient interdependent self-construal tend to prefer interpersonally oriented advertisements with messages, such as “making your friends and family proud” [61,62]. Hesapci et al. [58] also found individuals with an interdependent self-construal display had more positive evaluations toward an in-group ethnic ad model than did individuals with independent self-construal. In addition, high-RISCs are more likely to pay attention to information about others’ relationship, thus high-RISCs would be more likely than low-RISCs to pay attention to social relationship cues in advertising [26]. Martin et al. [26] also found that people who define themselves with high-RISCs respond most favorably to advertisements featuring a dyadic relationship (two people), and in contrast, people with low-RISCs respond most favorably to solitary models. From the above, we can know people with high-RISC would cherish the relationships more between family and friends than those with low-RISC. We proposed the following hypothesis:

**Hypothesis** **3.**RISC and social relationship cues interact to affect behavioral intention.

## 3. Study 1

### 3.1. Method

The present research used two studies to demonstrate the effects of anti-smoking advertisements. The duration of the experiment process of Study 1 and Study 2 for each participant was 10–15 min, including the exposure to the print anti-smoking advertising and completing the questionnaire. For ensuring consistency in paying attention to the ad exposure and processing the contents, we assigned the same assistant to contact participants, describe the experiment process, and oversee the completion of all experiments. Study 1 examined the interaction effect of self-referencing and advertising appeal toward the advertisement.

#### 3.1.1. Experiment Design

Study 1 employed a 2 (self-referencing: narrative vs. analytical) by 2 (advertising appeal: rational vs. emotional) factorial design (see Appendix B), where the self-referencing and advertising appeal act as the measured independent variables and behavioral intention as the dependent variable.

#### 3.1.2. Sampling

We selected one of the biggest senior high schools in Taiwan and employed convenience sampling, with students in this school as the unit of analysis. We designed four versions of advertising, and the participants needed to complete the questionnaire after seeing one version of the advertisement. In addition to age and gender, we also measured “having a smoking habit?” by asking the participants “Do you/your friends/your family smoke?” (1 = yes and 2 = no). After completing the survey, each participant received a $1.50 gift card for participating. A total of 192 respondents (62 males and 130 females) completed a questionnaire in Chinese. The profiles of the respondents are shown in Table 1. The questionnaire used to collect data contained an experimental design. The results showed no significant difference exists between males and females in the behavioral intention of anti-smoking (*p* = 0.093). There were also no age differences in participants’ reaction to the ads (*p* = 0.237).

#### 3.1.3. Measurements

In Study 1 and 2, all the measures used 7-point scales. We adopted the method most studies use to combine the measures and take the mean, and then followed three steps to complete the Chinese translation of the questionnaire items to ensure the validity of the instruments. First, two authors translated the questionnaire items separately, and then the differences in questionnaire translation were discussed together. Second, two authors referenced the Chinese translation of items from a local researcher and discussed and revised the content. Third, if there were still disagreements, the third author was queried to determine the final version of the Chinese instruments. To assess behavioral intention of anti-smoking, we used the scale from Martin et al. [26], including “I am now less likely to smoke than I was before seeing that ad” and “I am now more interested in learning about the consequences of smoking than I was before seeing the ad.” (see Appendix A).

In the analytical self-referencing, ad messages directly address participants with second person pronouns (e.g., “you”) and talks about how smoking will hurt “your” health and influence others. On the other hand, this study manipulates narrative self-referencing by asking participants to imagine themselves smoking and how smoking would harm their health. To assess the degree of self-referencing, participants were asked to complete two items: “The ad related to me personally” and “To what extent did your thoughts focus on you personally?” [38].

#### 3.1.4. Reliability and Manipulation Check

We used Cronbach’s α to analyze whether the scales for all of the constructs had acceptable levels of reliability [63]. Cronbach’s α values of behavioral intention (0.70) reached the threshold of 0.7, indicating the construct had acceptable reliability.

The manipulation check defined by Hoewe [64] is “a test used to determine the effectiveness of a manipulation in the experimental design.” If the manipulation check is successful (that is, the expected difference between or among experimental conditions), the researcher can satisfactorily conclude participants correctly understand, interpret, or respond to the stimulus and draw more accurate conclusions about the relationship between the independent and dependent variables. We performed manipulation checks to confirm the experiment manipulations were successful. In the emotional advertising appeal, the present study aimed at evoking feelings or emotions in participants, and rational appeal focused on the benefit of not smoking to the participants, such as health and economy. To test the effectiveness of our manipulation, after seeing the advertisement, participants were asked to respond to the following two rational items: “This advertisement contained a lot of rational information”; “This advertisement mainly conveys the possible harms and consequences of smoking to the body through rational appeals” and two emotional items: “This advertisement has a very strong appeal to my emotions”; “This advertisement mainly conveys the probable distress and consequences of smoking for family and friends through humanistic and emotional appeals” [33]. The ANOVA results indicated a significant difference between emotional and rational advertising appeal (rational appeal, *p* = 0.044; emotional appeal, *p* = 0.021). The results indicated the manipulations were as successful as intended.

### 3.2. Results: Statistical Analysis

ANOVA provides a method of data analysis that is based on the consideration of experimental design [65]. It is the most efficient method that can be used to analyze experimental data [66]. Thus, to analyze the main effect of each variable in detail, we conducted ANOVA to test the hypotheses. ANOVA tests the statistical significance of the difference in means (central tendency) among different groups of scores. Different groups of scores may correspond to different levels of a single independent variable or different combinations of levels of two or more independent variables [67]. The means for behavior intention are 5.62 ± 0.10 (narrative self-referencing), 5.39 ± 0.10 (analytical self-referencing), 5.59 ± 0.11 (rational advertising appeal), and 5.42 ± 0.10 (emotional advertising appeal). The ANOVA results (Table 2) showed that the different levels of self-referencing (*p* value = 0.796) and advertising appeal (*p* value = 0.186) do not significantly affect behavior intention. However, the results revealed a significant two-way interaction between self-referencing and advertising appeal on behavior intention. Since an interaction effect had emerged for the behavior intention, we carried out a simple main effect test for self-referencing and advertising appeal to examine the differences in greater detail.

All participants were randomly assigned to one of the four experiment conditions, and they could leave at any time if they did not want to participate in the experiment (N_AE_ = 52, N_AR_ = 50, N_NE_ = 53, and N_NR_ = 36, individually; see Table 3). We used ANOVA to demonstrate the simple interaction effect. The results showed that when participants with analytical self-referencing processing watched the rational appealing advertising, it would cause more behavior intention than when they watched emotional appealing advertising. However, if participants had a narrative self-referencing process, watching emotional appealing advertising led to higher behavior intention than watching rational appealing advertising, so the results supported Hypotheses 1 and 2.

## 4. Study 2

### 4.1. Method

Study 2 examined the following ad effect: (1) the interaction effect between RISC and social relationship cue; (2) the interaction effect between self-referencing, RISC, and social relationship cue; and (3) the mediating effect of self-referencing on the relationship between social relationship cues and behavioral intentions.

#### 4.1.1. Experiment Design

Study 2 employed a 2 (RISC: high and low) by 3 (social relational cue: self, friends, and family) factorial design (see Appendix C), where the social relational cue and RISC act as the measured independent variables and behavioral intention as the dependent variable. In addition, RISC used the median split method to divide into high and low. The research used ANOVA and regression to examine the data.

#### 4.1.2. Sampling

We selected the same senior high school as Study 1 and employed convenience sampling, with students in this school as the unit of analysis. We designed three versions of the advertisement, and the participants needed to complete the questionnaire after seeing one version of the advertisement. After finishing the survey, every participant received a $1.50 gift card for participating in this survey. A total of 222 respondents (91 males and 131 females) completed a questionnaire in Chinese. The profiles of the respondents are shown in Table 4. The questionnaire used to collect data contained an experimental design. The results showed no significant difference between males and females for behavioral intention of anti-smoking (*p* = 0.153). There were also no age differences in participants’ reaction to the ads (*p* = 0.096).

#### 4.1.3. Measurements

In Study 2, participants evaluated the behavioral intention items from Martin et al. [26] as in Study 1. RISC was measured using the 11-item scale of Cross et al. [21], including statements such as “My close relationships are an important reflection of who I am.”

#### 4.1.4. Reliability and Manipulation Check

We also used Cronbach’s α in Study 2 to determine whether the scales of the constructs had acceptable levels of reliability [59]. The Cronbach’s α values of behavioral intention (0.71) were higher than 0.7, indicating the construct had acceptable reliability.

### 4.2. Results: Statistical Analysis

To analyze the main effect of each variable in detail, we conducted ANOVA to test the hypotheses. The means for behavior intention were 4.51 ± 0.189 (social relationship cue_self; N_self_ = 76), 5.01 ± 0.190 (social relationship cue_family; N_Family_ = 77), 4.35 ± 0.223 (social relationship cue_friends; N_Friends_ = 69), 5.13 ± 0.170 (RISC_high), and 4.19 ± 0.15 (RISC_low). The ANOVA results showed that the different levels of social relationship cues (*p* value = 0.306) did not significantly affect behavior intention (Table 5). In addition, the results also revealed that behavior intention did not be significantly affected by the two-way interaction between social relationship cues and RISC, so the results did not support Hypothesis 3. From the means of the six cells (Table 6), we can determine whether the participant has high RISC; the advertisement with higher social relationship cues will raise his (her) behavior intention of anti-smoking. On the other hand, if the participant is low RISC, family also had higher influence on him/her than the other two relationship cues, that is to say, even if the participant is an independent person, he/she would also follow his/her family’s advice on something, including health problems. The results also showed that RISC significantly affected behavior and that high-RISC participants had higher behavior intention than low-RISC ones.

## 5. General Discussion

Based on a report from WHO [68], more than 7 million people die each year from tobacco-related diseases. As smoking has emerged as a health-risk behavior, scholars and practitioners have put a lot of effort into finding ways to quit smoking [69]. Globally, the vast majority of anti-smoking messages are based on appeals about the negative effects of fear on the (potential) smoker himself/herself. Miller et al. [70] suggested that such a global strategy might be suboptimal. This research examined (1) how self-referencing and advertising appeals interact to affect anti-smoking intention and (2) how social relationship cues and RISC interact to affect anti-smoking intention. The results show that, for emotional appeal advertising, narrative self-referencing leads to better behavioral intention than analytical self-referencing. On the other hand, for rational appeal advertising, analytical self-referencing leads to better behavioral intention than narrative self-referencing. Study 2 used social relationship cues and RISC to find the interaction effect on participants’ behavior intention of anti-smoking after seeing the advertisement. Even though the results showed these two constructs had no interaction effect, the results still showed that people with interdependent RISC would have higher behavior intention of anti-smoking than those with independent RISC.

### 5.1. Theoretical Contributions

This research makes several theoretical contributions. First, few studies have used different approaches, such as psychology or social relationships, to explore high school students’ reaction to an anti-smoking advertisement. The present research used two studies to fill the research gap by focusing on how advertising appeal, self-referencing, and RISC influence the participants’ behavior intention when they see an anti-smoking advertisement.

Second, most anti-smoking advertisements use rational or emotional appeal to persuade smokers to quit smoking. However, they do not seem to work very well. Therefore, the present study manipulated self-referencing and advertising appeal to demonstrate how these two constructs affect anti-smoking intention.

Third, this research revealed self-referencing as a moderator of anti-smoking intention for different appeals. This research was also based on previous research (e.g., Martin et al. [26]; Meyers-Levy and Peracchio [14]), suggesting the possibility of inducing self-referencing through the design of an anti-smoking ad.

Fourth, this research also contributes to current anti-smoking advertising research, suggesting insights into ad effectiveness can be attained by considering the personal aspects of the audience. For example, Christie et al. [71] speculated individual differences offer a useful perspective for research on alcohol consumption. Likewise, this research shows that RISC is a key individual difference influencing evaluations of anti-smoking advertising. Further, the anti-smoking intentions can be influenced if RISC is considered. Thus, scholars researching anti-smoking advertising should consider the role of RISC when developing anti-smoking advertising communication.

### 5.2. Managerial Implications

The results of this research showed that, if the anti-smoking advertisement used rational appeal, it can use more analytical self-referencing content to persuade viewers, because analytical self-referencing persuades through dual cognitive response processes [40]. The results correspond to the statement of Belch and Belch [72], who said consumers process rational persuasion using facts, logic, and analytical and sequential thinking [73]. On the other hand, if the anti-smoking marketers implement an emotional strategy to make the participants believe the opinions in the advertisements, they can also use narrative self-referencing together with it to make the target audience more immersed in the advertisements. In addition, under conditions of narrative transportation, affective responses influence persuasion rather than the systematic analysis of message strength [40]. The results are also in line with the research of Block [74], which stated self-reference ads are especially effective when evoking negative emotions, such as fear and guilt [75]. Smokers who experienced more transportation to anti-smoking messages reported it would be hard to quit smoking, and the effect was mediated by both experiential (emotional and self-reference) and cognitive response to the messages [76]. Thus, when using “you” statements in these kinds of advertisements, the more an individual relates an ad to himself/herself, the greater the likelihood of recall and favorable evaluation [77].

Study 2 used social relationship cues and RISC to find the interaction effect on participants’ behavior intention of anti-smoking after seeing the advertisement. Even though the results showed these two constructs had no interaction effect, the results still showed that people with interdependent RISC would have higher behavior intention of anti-smoking than those with independent RISC. This is because people possessing an interdependent self-construal show a higher level of association with others and a lower level of distinction from others [53,78]. However, from the results we found that even young participants with independent RISC predominantly rely on what their family said to them. Martin et al. [26] found low-RISCs evaluate the ads featuring solitary models more favorably than the ads featuring a dyadic relationship (two people), and vice versa. Thus, based on the “self-congruency proposition,” individuals with a strong independent self-construal prefer messages emphasizing the personal aspects of an entity, but individuals with a strong interdependent self-construal prefer messages emphasizing the relational aspects of an entity [79]. Thus, to consumers with a culturally salient independence as compared to interdependent self-construal, advertisements focusing on individual benefits and preferences are more effective [80].

### 5.3. Limitations and Future Research

The present study only used selected psychology variables, such as self-reference and RISC, to investigate the effect on participants’ behavior intention of anti-smoking. Future research can use other psychology variables, such as self-affirmation or empathy, to understand how the psychology variables moderate the effect of anti-smoking advertisements.

The present study only examined one group of senior high school students in Taiwan. Future studies can expand the scope to include other research subjects, which will give the results greater significance in demonstrating the effect of anti-smoking advertising. Different types of high school (e.g., vocational high schools and general high schools) or academic institution (e.g., colleges, universities) might generate different effects on the evaluations of anti-smoking advertisements, and thus the results may differ.

Future study could explore the roles of culture and social norms. Some elements of culture and social norms may also affect the effectiveness of anti-smoking advertisements, such as individualism–collectivism and self-construal. Our research results showed that the effect of advertising with family as a social relationship cue was better than that with peers and friends. One possible reason could be our experiment participants are of Chinese ethnicity and cultural background. The Chinese admire collectivism more than individualism [81], so even though they are teenagers, they may still care more about the reactions of their family than that of their friends. In addition, it is also possible that since the subjects live together with family members, if someone smokes, their family will be more affected than will their peers, so they will pay more attention to family members’ reactions. Furthermore, research has shown people with different self-construals have their own unique notions, feelings, and behavior [53]. Even though people may live in the same environment, self-construal also leads people to develop distinct perceptions [82], processes of decision-making, and selection [83].

This study used convenience sampling, which did not allow us to control the smoking rate of the sample and thus might result in limitations of result interpretation. Although the smoking rate of high school students in Taiwan was originally not high, only 8.4% [84], which is not much different from the smoking rate of Study 1 (5.7%), future research could focus solely on smokers to explore the impact of anti-smoking advertisements on them.

## 6. Conclusions

Although smokers are more likely to get many diseases than nonsmokers [85], they continue to smoke. In addition, the adolescents’ smoking intentions are stronger than before especially for the e-cigarettes [86]. The present study used the multidisciplinary concepts from psychology and sociology to investigate the effects of anti-smoking advertising on adolescents’ smoking intention.

The results showed that rational appealing advertising is suitable for analytical self-referencing and emotional appealing advertising is suitable for narrative self-referencing. In addition, when the anti-smoking advertising is from the family perspective to persuade adolescents not to smoke, both influence the adolescent more than the other two social relationship cues (self and friend).

In terms of academic contribution, this research results fulfill the research gap of anti-smoking advertising on the psychological and sociological level. In terms of practical contribution, this research hopes to inform relevant units that which copywriting and appeals can be used to make the target group more acceptable when using anti-smoking advertising.

## Figures and Tables

**Table 1 ijerph-17-08481-t001:** Description of respondents in Study 1.

Item	Description	Frequency	Percentage
Gender	Male	62	32.3%
	Female	130	67.7%
Age	16	70	36.4%
	17	92	47.9%
	18	28	14.6%
	19	2	1%
Have a smoking habit	Yes	11	5.7%
	No	181	94.3%
Friends have a smoking habit	Yes	92	47.9%
	No	100	52.1%
Family has a smoking habit	Yes	117	60.9%
	No	75	39.1%

**Table 2 ijerph-17-08481-t002:** ANOVA results of self-reference and advertising appeal on behavior intention.

Source	Behavior Intention		
	df	F	*p*
Self-referencing	1	0.067	0.796
Advertising appeal	1	1.759	0.186
Self-referencing × Advertising appeal	1	8.620	0.040 **

** Deemed significant at the 0.05 level.

**Table 3 ijerph-17-08481-t003:** Simple main effect of self-referencing.

Self-Referencing	Advertising Appeal		*p*
	Emotional	Rational	
**Analytical**	5.14 (*n* = 52)	5.63 (*n* = 50)	0.022 **
**Narrative**	5.80 (*n* = 53)	5.38 (*n* = 36)	0.067 *

* Deemed significant at the 0.1 level; ** deemed significant at the 0.05 level.

**Table 4 ijerph-17-08481-t004:** Description of respondents in Study 2.

Item	Description	Frequency	Percentage
Gender	Male	91	39%
	Female	131	61%
Age	16	30	13.5%
	17	87	39.2%
	18	66	29.7%
	19	15	6.8%
	20	4	1.8%
	>21	20	9.0%
Have a smoking habit	Yes	68	30.8%
	No	154	69.2%
Friends have a smoking habit	Yes	165	74%
	No	57	26%
Family has a smoking habit	Yes	141	63.3%
	No	81	36.7%

**Table 5 ijerph-17-08481-t005:** ANOVA results of social relationship cues and relational-interdependent self-construal (RISC) on behavior intention.

Source	Behavior Intention		
	df	F	*p*
Social relationship cue	2	1.189	0.306
RISC	1	13.354	<0.001 ***
Social relationship cues × RISC	2	0.533	0.649

*** Deemed significant at the 0.001 level.

**Table 6 ijerph-17-08481-t006:** The means of cells.

RISC	Social Relationship Cue		
	Self (*n* = 76)	Family (*n* = 77)	Friends *(n* = 69)
High RISC	4.83	5.47	4.97
Low RISC	4.23	4.35	4.10

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
