# Peer review of "The Moderating Effects of Self-Referencing and Relational-Interdependent Self-Construal in Anti-Smoking Advertising for Adolescents"

_ijerph, 2020, doi:10.3390/ijerph17228481_

Round 1

Reviewer 1 Report

Thank you to the authors for providing comprehensive responses to my previous comments. I have a few additional concerns/comments, mainly concerns relating to the change in hypotheses and also instances where I felt that changes to the manuscript were not sufficiently made after the previous round of comments.

MAJOR COMMENTS

It is not acceptable to change hypotheses between submissions, and this leads me to be concerned about HARKing and whether the initial hypotheses were actually generated prior to collecting/analysing data. The authors also did not pre-register their analyses from what I can see, and so we have no idea if these hypotheses were formulated prior to data collection/analysis. I’m aware that the authors revised the hypotheses in response to my comments about not understanding a phrase, but rather than clarifying their terms the authors changed the whole hypothesis/dropped outcomes?? To overcome these concerns, the authors must frame their analyses as exploratory without hypotheses.

The sampling strategy and participant selection criteria are still not sufficiently described. Were there any selection criteria for ppts? If so, what were they? The response given in the previous response to reviewers does not answer these questions.

MINOR COMMENTS

Comment #21 in my last round of comments “Studies 1 and 2, sampling/measurements – please state how you measured ‘have a smoking habit’?” – the authors responded that they asked the participants “Do you/your friends/family smoke?” 1=yes and 2=no – but this is not a measurement of the participants’ own smoking habit? Please clarify.

I believe that in several instances changes to the manuscript should have been made as well, to make the manuscript clearer for other readers:

  1. Comment #6 in my last round of comments: Please define and/or provide examples of the following in the manuscript text: "gain or loss-framed messages", “appealed to fear such as death or anger to examine the anti-smoking effect”
  2. Comment #7 in my last round of comments: Please describe in the manuscript in greater detail what you mean by “explicit reminders of death fail to produce the main effects on health-related behaviors and attitudes” – what are the main effects?
  3. Comment #12 in my last round of comments: Please define advertising effect in the manuscript.
  4. Comment #15 in my last round of comments: Please define behavioural intention in the hypotheses in the manuscript.
  5. Comment #20 in my last round of comments: Please state that the mean was taken in the manuscript.
  6. Comment #21 in my last round of comments: Please state this measurement in the manuscript.
  7. Comment #23 in my last round of comments: Please describe the manipulation check in the manuscript.

Reviewer 2 Report

This new revised version of the manuscript has significantly improved and several of previously noted comments have been addressed. More specifically the authors have described different forms of smoking and cited important literature relevant to tobacco control and advertisement.  In the discussion, several important citations have been added and the theoretical contributions of the study have been more clearly described.

Some other shortcomings, however, have not yet been addressed in the revised version. My main concerns are related to the study design and the results section. In the study design, it has not been described how "having a smoking habit" has been measured. Furthermore, considering the addictive nature of tobacco use, it is not clear how behavioral intention in someone with a smoking habit can be defined. Lastly, no information are provided related to the following questions:

  • How participants were exposed to the ads (i.e., for how long, where, when)? How the exposure was standardized to ensure consistency in paying attention to and processing the contents)  
  • How the instruments were translated into Chinese and validated?
  • What would be the role of culture and other environmental norms?
  • Was there any gender or age differences in participants' reaction to the ads?

Reviewer 3 Report

It's a good topic and it maybe useful to offer administrative suggestion in the rule of tobacco ads.

However, some of the sentences are too long to be easily understood.

And the sample size is relatively small, especially the smoking proportion is quite low. So the interpretation of the results maybe more prudent.

Round 2

Reviewer 1 Report

Thank you to the authors for revising their manuscript in such a quick time frame. I believe this manuscript is now OK to be published. There is still a lot of jargon and it would benefit from being checked / improved in terms of English language, but methodologically it is acceptable. The authors mentioned that they hired another native-speaking academic editor to assist with grammar, syntax, and style, which is commendable, but there is still a great amount that can be improved. It's currently very difficult to read as it stands.

Two other minor points:

Abstract

Could the authors please include some numbers (%s and statistical coefficients) to support the results described in the manuscript, rather than simply giving a narrative overview?

Comment #11 in my last round of comments: Please describe the manipulation check in the manuscript

The authors added a definition of a manipulation check, but in my last set of comments I meant that I would like to see a description of the specific manipulation check that the authors themselves performed, and why, reported in the manuscript.

Reviewer 2 Report

I am pleased with the authors' responses and the new changes to the manuscript 

Reviewer 3 Report

Some of the percentage in table1 and table4 are not correct. Please check all your data in the tables and clearly mark the row or column descriptions. 

Author Response

This manuscript is a resubmission of an earlier submission. The following is a list of the peer review reports and author responses from that submission.

Round 1

Reviewer 1 Report

The literature review is well-written and concepts of self-referencing are clearly defined, although there are improvements that should be made which I suggest below. The methods are not clear and need to be greatly improved. The results are brief, but given the hypotheses and design of the study this is OK. The discussion is appropriate given the findings and limitations of the study design. However, the topic of this paper is quite niche with more implications for theory than for public health; I’m therefore not entirely convinced it would appeal to the audience of IJERPH and may be better suited to a psychology journal that is theory-based.

Introduction/literature review – I don’t think it’s conventional to have a separate introduction and literature review in this type of publication. Have the authors considered just having an introduction, which integrates the literature review (i.e., without a separate section)? This would streamline the publication.

Introduction – the second paragraph of the introduction covers some important points but the language used is jumbled, with lots of jargon, and it’s not really clear what the previous literature found or what the gaps are. Could the authors revise this paragraph to be clearer? Some questions/suggestions:

  1. Can you please define and/or provide examples of ‘gain or loss-framed messages’ and also expand on what you mean by …appealed to fear such as death or anger to examine the antismoking effect’ – I don’t really understand what this means
  2. ‘However, some research suggests explicit reminders of death fail to produce the main effects on health-related behaviors and attitudes’ – what are these main effects?
  3. ‘Meyers-Levy and Peracchio’ – is this a typo? Should there be a full stop before this?
  4. What is a ‘salutary effect’?
  5. ‘The positive effect of self-referencing on ad and brand judgement…’ – this is the first time brands are mentioned, do you mean brands of cigarettes? What is brand judgement? I don’t understand how self-referencing would influence what brands to choose, if that’s what you mean. Perhaps have a separate paragraph discussing the relation between self-referencing and brand ‘judgement’ (whatever that is) or consider dropping completely since I don’t see how it’s relevant to your study
  6. What is ‘the advertising effect’?

Literature review, advertising appeal – this section seems quite broad. Surely there is specific literature of the impact of cigarette advertising/appeal, and/or anti-smoking advertising/appeal? It would be best to include this if possible. Similarly, is there any research about self-referencing and cigarette advertising specifically? If so, please incorporate it. If not, please state there is none (that you’re aware of).

Hypotheses – these would be best placed at the very end of the literature review, rather than scattered throughout. They also need to be revised to be more specific and falsifiable:

  • All hypotheses: ‘Behavioural intention’ to what? To smoke (ever or regularly)? To quit smoking? ‘Advertising appeals’ of what? Advertising of cigarettes? Anti-smoking advertising?
  • H1a/b: What does ‘will be persuasive’ mean? How will this be tested? Hypotheses are usually something like ‘X will be associated with Y’, or ‘X will increase Y’
  • H3: ‘Evaluations’ of what?

Studies 1 and 2 – you have not described anywhere what the participants were actually exposed to. What did the antismoking advertisements consist of? Were they videos? Images? Posters? Was there writing or images only, or both? How long were they shown the adverts for? Perhaps you could provide some examples using pictures.

Studies 1 and 2, measurements – could you just state that two measures were used for behavioural intention, and write these out? (‘I am now less likely to smoke than I was before seeing that ad’ and ‘I am known* more interested in learning about the consequences of smoking than I was before seeing the ad’) *You have a typo here. This is important for interpreting the findings. Please also state the response options, either in the main text or Appendix. Also, being interested in learning about the consequences of smoking is quite different from being less likely to smoke; what is your justification for combining these measures? How correlated were they? If not, consider whether it’s really appropriate to combine them.

Studies 1 and 2, measurements – please state how the measures were combined, e.g., the two behavioural intention measures. Were they combined and the mean taken (and hence was the final score a 7-point scale), or were they summed? The same goes for the other measures.

Studies 1 and 2, sampling/measurements – please state how you measured ‘have a smoking habit’?

Studies 1 and 2, sampling – the sampling strategy and participant selection needs to be greatly expanded. Did you have any participant selection criteria? Were there any missing data?

Studies 1 and 2, reliability and manipulation check – what manipulations? This is the only mention of any manipulations, please specify/elaborate. If you mean manipulation of your independent variable (rational/emotional appeal), then state this explicitly, although this will also need some explaining as to why manipulation checks are necessary?

Pg 1 line 35 – please define the acronym ‘CDC’

Reviewer 2 Report

This is an interesting study that aims at testing three hypotheses related to self-referencing, advertising appeal, relationship independent self-construal, and behavioral intention. Overall, the authors have done a good job in positioning their hypotheses and designing two case studies to test them. The research, however, lacks details in the following areas and the quality of argument can be further improved:

  • The title is misleading and is not representative of the study aims and the findings
  • The introduction is too long and somehow more focused on relations between a few constructs of interest rather than on providing a comprehensive discussion regarding youth tobacco use problem and the gaps in the literature. Most examples were from commercial world with primary objectives of convincing the adoption of new products/services rather than fostering healthy behaviors.
  • The methodology section has been mixed with findings from two case studies and lack specific details related to the contents of the four versions of the advertisement, the process of assigning different categories, the length of exposure to the intervention, etc. Furthermore, the study constructs such as having a smoking habit has been loosely defined. It is not clear whether the authors meant regular use, and if so, how frequent/severe. Friends and families having smoking habits are also too general. This section needs major revision.
  • The final argument can be further improved and strengthened. It is not clear how the findings add to the literature.
  • There is no mention of ethical considerations and IRB approval